# Synthesis, Characterization and Photocatalytic Activity of CoFe_2_O_4_/Fe_2_O_3_ Dispersed in Mesoporous KIT-6

**DOI:** 10.3390/nano12203566

**Published:** 2022-10-12

**Authors:** Johnatan de Oliveira Soares, Wesley Eulálio Cabral Cavalcanti, Marco Antonio Morales Torres, Sibele Berenice Castella Pergher, Fernando José Volpi Eusébio De Oliveira, Tiago Pinheiro Braga

**Affiliations:** 1Instituto de Química Laboratório de Peneiras Moleculares (LABPEMOL), Instituto de Química, Universidade Federal do Rio Grande do Norte, Natal 59078-970, RN, Brazil; 2Departamento de Física Teórica e Experimental, Universidade Federal do Rio Grande do Norte, Natal 59078-970, RN, Brazil

**Keywords:** CoFe_2_O_4_, Fe_2_O_3_, ferrite, KIT-6, photocatalysis

## Abstract

The present work aimed to synthesize and characterize a solid based on CoFe_2_O_4_/Fe_2_O_3_-KIT-6 and evaluate its performance in the photocatalytic degradation of the remazol red ultra RGB dye. By analyzing XRD, N_2_ physisorption, and Mössbauer results, it was possible to identify that the desired CoFe_2_O_4_/Fe_2_O_3_ phase was achieved, which maintained its structural properties. The FTIR-pyridine indicated the presence of Lewis acid sites, while TPD-CO_2_ showed a large amount of weak basic sites. The band-gap energy indicated that the compound can be applied in photocatalytic degradation under UV/visible light, with the possibility of magnetic separation at the end of the reaction. The photocatalysis results indicated that there was complete degradation of the remazol red ultra RGB dye within 1 h of reaction. Despite the absence of H_2_O_2_, the combination of the proposed photocatalyst with the anatase phase (TiO_2_) showed significant improvements in the degradation process. The proposed mechanism for complete dye degradation indicated that a sequence of radical reactions is necessary, generating oxidant species such as •OH and the final products were CO_2_ and H_2_O.

## 1. Introduction

Several harmful organic substances such as industrial dyes, pharmaceuticals, agrochemicals, and oil refinery residues are widely detected in wastewater, causing environmental problems in several sectors of society.

Furthermore, these contaminants are released into nature through domestic sewage, and by industries such as leather, paper, plastic, pharmaceutical, textile, and food. These contaminants are chemically stable species, toxic, and they cause serious problems to the ecosystem and human health, even at low concentrations [1,2].

Dyes are a class of contaminants that stand out, since they are the main polluting agents of the aquatic environment [3]. This is because of their intense color and very heterogeneous chemical composition, which can affect the natural biochemical cycles of aquatic environments and make it difficult to carry out photosynthesis, leading to the loss of biodiversity [4]. The presence of dyes that are not removable by conventional coagulation, sedimentation, and filtration processes are harmful for the environment. Direct and long-term contact with dyes can cause allergies and respiratory problems; in addition, certain classes of dyes can lead to carcinogenic and mutagenic conditions [5,6,7].

Thus, the need to remove pollutants such as industrial dyes from effluents before releasing them into aquatic systems is of the utmost importance. For this purpose, there are several methods that allow their removal and/or degradation such as adsorption, sedimentation, coagulation, flocculation, chemical precipitation, membrane filtration, distillation, ion exchange, extraction, crystallization, ultrafiltration, reverse osmosis, ozonolysis, electrodialysis, electrolysis, and advanced oxidative processes (AOPs) such as Fenton [8].

Among them, AOPs stand out for using oxidizing agents, such as hydroxyl radicals (HO•), peroxyl radicals (•O_2_-), and hydroperoxide radicals (HOO•), to degrade and mineralize organic materials into carbon dioxide and water [9]. These processes can be divided into two groups: firstly, degradation occurs in the presence of irradiation (photocatalytic processes); secondly, oxidation occurs by electrochemical processes [10]. Photocatalytic processes are the solution with the highest potential for total mineralization of organic compounds in wastewater; they use several materials such as TiO_2_, CeO_2_, and iron-based solids. Furthermore, several transition metal oxide semiconductor photocatalysts have been investigated due to their attractive properties such as high electron mobility, low toxicity, chemical stability, and easier separation compared with homogeneous photocatalysts [11,12,13]. Specifically, some studies show different photocatalysts in the degradation of red remazol dyes; however, there is a lack of studies on materials such as espinels ferrites. It is worth pointing out that remazol dyes are comparatively less reactive than other dyes, making them more stable in water and more difficult to degrade [14,15,16,17].

In this context, espinels ferrites stand out for being a wide class of highly stable magnetic oxides containing iron in their structure and can be produced by many synthesis routes [18]. Among the various applications of ferrites, it is possible to mention the generation of hydrogen gas through the water splitting, synthesis of compounds of high industrial value such as styrene, in piezoelectric and magnetic components, biomedical applications, photoanodes, photocathodes, and in the degradation of toxic substances present in the environment from industrial and urban processes [18].

The magnetic properties of ferrite compounds are suitable for the reusability of heterogeneous catalysts, since they greatly simplify the separation operations and the reuse of catalysts by the application of an external magnetic field, which facilitates their recovery after the reaction process. In addition, ferrites have excellent absorption properties in the visible spectrum due to their low band gap energy related to the electron-hole recombination rate [9,19,20]. The ferrites, when combined with other photocatalysts, such as anatase (TiO_2_), may provide advantages in terms of activation at wavelengths above 400 nm, as well as facilitating separation at the end of the process. On the other hand, the separation of pure TiO_2_ from polluted wastewater still remains a challenge [21,22]. Some studies have already shown the successful use of ferrites combined with other oxides in the degradation of organic matrices [23,24].

It is also possible to improve the properties of ferrites and other photocatalysts by combining them with porous supports such as mesoporous silicas and molecular sieves, which facilitate the separation and allow an increase in the surface area, the dispersion, and exposure of the active sites, preventing their agglomeration [25,26].

KIT-6 is one of the most used supports in catalysis. This type of mesoporous silica has regular pore size; a well-ordered type Ia3d three-dimensional cubic mesophase; a continuous, interpenetrating network of channels; a high surface area; a large pore volume; and a chemically modifiable surface through functionalization processes [27]. Despite being widely applied in catalysis and adsorption, there are few studies involving KIT-6 and other silica-based mesoporous materials as catalytic supports for reactions involving ferrites and photocatalysis.

Thus, systems combining cobalt ferrite with other active phases, such as Fe_2_O_3_, dispersed in porous supports such as KIT-6 can be an interesting alternative for the elimination of dyes. Therefore, the present work described the synthesis of CoFe_2_O_4_/Fe_2_O_3_ oxides supported on a KIT-6 mesoporous matrix. It also evaluated its morphological, structural, and chemical characteristics, and its performance in the photocatalytic degradation of the industrial dye remazol red ultra RGB.

## 2. Materials and Methods

### 2.1. Synthesis of KIT-6 Support

The synthesis of KIT-6 considers the reactants in the molar proportion of 1.00 TEOS; 0.017 P123; 1.83 HCl; 195 H_2_O; and 1.31 Butanol [28]. The organic director P123 was dissolved in water and HCl (37% *w*/*v*) under stirring for 3 h at 35 °C. Then, butanol was added and the system remained for approximately 1 h at the same temperature and under constant magnetic stirring. Afterward, TEOS was added and the gel was maintained under stirring at 35 °C for 24 h.

The gel was placed in an autoclave for hydrothermal treatment under static conditions at 100 °C for 24 h. After that, the material was rinsed several times with an ethanol solution containing 2% HCl until reaching pH = 7. Finally, the solid was calcined at a temperature of 550 °C at a heating rate of 1 °C min^−1^ for 6 h under an air atmosphere. The synthesized solid was named KIT-6.

### 2.2. Synthesis of Catalysts

The synthesis of 20% by mass of the CoFe_2_O_4_/Fe_2_O_3_ catalyst supported on KIT-6 was carried out in order to obtain 1 g of the final catalyst by the classical incipient impregnation method. Initially, an aqueous mixture containing iron nitrate (Fe(NO_3_)_3_·9H_2_O) and cobalt nitrate (Co(NO_3_)_2_·6H_2_O) was dropped on the KIT-6 support until the wet point was reached. Then, the material was dried at 100 °C. This procedure was repeated until the solution containing Fe^3+^ and Co^2+^ metals was completely used. Subsequently, the sample was calcined in air at 700 °C for 2 h using a heating rate of 10 °C min^−1^. The solid formed was designated as CoFe_2_O_4_/Fe_2_O_3_-KIT-6.

The same methodology aforementioned was used to modify the photocatalyst CoFe_2_O_4_/Fe_2_O_3_ -KIT-6 with TiO_2_. Titanium isopropoxide, 10% by mass of anatase, was added under the CoFe_2_O_4_/Fe_2_O_3_-KIT-6 catalyst. The amounts of reactants were calculated in order to obtain 1 g of solid. Titanium isopropoxide was dissolved in anhydrous ethyl alcohol and dripped onto the catalyst until reaching the wet point. Posteriorly, the material was placed in an oven at 100 °C for 20 min. This procedure was repeated until all the precursor solution was completely used. Then, the sample was calcined in air for 3 h at 570 °C and at a heating rate of 10 °C/min^−1^; the obtained material was designated as TiO_2_/CoFe_2_O_4_/Fe_2_O_3_ -KIT-6.

### 2.3. Characterizations

The X-ray diffractograms (Hardtstraße, Karlsruhe, Germany) were obtained in a Bruker D2 Phaser Diffractometer equipped with CuKα radiation (λ = 1.54 Å) with a Ni filter, with steps of 0.02°, with angle 2θ ranging from 10° to 90°, to determine the cobalt ferrite and hematite crystal structure. Phase identification was performed using the X-Pert HighScore Panalytical software and the JCPDS-ICDD 2003 database. The Rietveld refinement was performed using GSAS-II software and EXPGUI interface (Los Alamos, NM, USA). The full width half-height (FWHM) of the peaks was used to calculate the crystallite size based on the Scherrer equation.

The Mössbauer ^57^Fe spectra were recorded in transmission mode using a spectrometer (SEECo) (Minneapolis, MN, USA) with a triangular speed scan. A closed cycle cryostat (Janis) was used to record spectra at 12 K and 300 K. The 14.4 keV γ radiation source had an activity of 20 mCi. The spectra were fitted using the software Normos90. Isomer shift values were relative to α-Fe recorded at 300 K.

The FTIR spectra of the materials were recorded with a Bruker Vector 22 spectrometer in the absorption mode with a resolution of 2 cm^−1^. The catalysts were placed in an IR Cell equipped with CaF_2_ windows and treated in situ. The different solids were first pretreated at 450 °C under vacuum for 1 h. The Pyridine was adsorbed under saturation vapor pressure for 5 min at room temperature, and the pellet was desorbed at 150 °C for 1 h to remove the physisorbed pyridine species. The sample spectrum after pretreatment and after pyridine adsorption were recorded in sequence to evaluate the acidic properties of the evaluated materials.

The temperature-programmed desorption of CO_2_ (TPD-CO_2_) was performed in the range of 40–500 °C under He flow (10 °C/min^−1^, 16 mL min^−1^). The samples were preheated under He flow (16 mL/min^−1^) at 700 °C for 1 h. Subsequently, the temperature decreased to 45 °C and the He flow was changed to pure CO_2_ (16 mL/min^−1^ for 0.5 h). The CO_2_ desorbed was identified by a thermal conductivity detector (TCD) after passing through a trap to remove any traces of water.

The N_2_ physisorption analyses were performed at a temperature of −196.15 °C in an ASAP 2020 physisorption/micromeritic equipment to obtain the textural properties of the catalyst. Prior to analysis, the solids were degassed under vacuum at 200 °C for 2 h.

The dispersion of cobalt/hematite ferrite was investigated by transmission electron microscopy (TEM) with an accelerating voltage of 120 kV (Jeol, JEM-2100, with EDS, Thermoscientific) (Tokyo, Japan). The solids were prepared by placing a drop of the ketone-solid dispersion on a carbon-coated copper support (300 mesh).

In order to determine the band gap energy of the catalysts, a UV-2450 SHIMADZU (Kyoto, Japan) spectrophotometer was used, in which the reflectance signals were collected in the range from 190 to 900 nm, with a resolution in the order of 0.5 nm. The analysis followed the standard barium sulfate powder method. The zeta potential was performed in a Stabino particle charge titration analyzer (Colloid matrix), the pH of which being adjusted throughout the measurements. Then, 40 μL of NaOH solution was added every 15 s until pH = 12 and an HCl solution was added until the pH reached 2.5. The created potential was detected and measured by electrodes, allowing the zeta potential to be obtained as a function of pH.

The magnetic characterizations were performed using a vibrating sample magnetometer (VSM) from Lakeshore at a temperature of 26 °C, with an external magnetic field ranging between −15 and 15 kOe.

### 2.4. Photocatalytic Tests

The photodegradation tests of the remazol red ultra RGB dye, from the company DyStar, were prepared using 100 mL of a 25 mg L^−1^ dye solution; 0.5 g of the catalyst was added for each liter of solution, and 2 mL of H_2_O_2_ with a concentration of 20 mmol/L^−1^. The pH value was adjusted to 3 by adding 1.0 mol/L^−1^ of H_2_SO_4_ after the addition of the photocatalyst. The system was kept under constant agitation for 2 h and irradiated with a 125 W mercury vapor lamp. Aliquots were taken at predefined reaction times for each test. The reaction was monitored by UV-VIS spectroscopy, with maximum absorbance at 516 nm. The aliquots taken for analysis were filtered with microporous filters. The solution was placed in a quartz-jacketed reactor connected to an electrostatic bath to keep the internal temperature of the reactor constant at 26 °C.

The solid was added to the solution and kept under constant magnetic stirring. Before starting the reaction, the solution was kept under magnetic stirring for 30 min in the absence of light to reach adsorption–desorption equilibrium. Subsequently, the lamp was turned on, initiating the reaction; 3 mL aliquots were collected using a syringe attached to a probe hose connected to the reactor core, filtered, and analyzed through absorbance readings from a UV-Vis spectrophotometer (DR5000-03, HACH).

Gas chromatography–mass spectrometry (GC-MS) measurements were performed to identify the major by-products from the degradation reaction of remazol red. The GC-MS is a GC-2010 Plus from Shimadzu with polar column (30 m × 0.25 mm × 0.25 mm) and with an automatic injection system from AOC-20i. The photocatalyst was separated from the ultra RGB red remazol dye (URRD) solution using 0.20 μm filters after 15 min of photodegradation. Then, the solutions were treated with a dichloromethane: URRD ratio of 2:1 and 1 mL (splitless) was injected. A heating ramp from 100 °C to 270 °C with a heating rate of 15 °C/min was used. The injector temperature was 270 °C. The NIST library (National Institute of Standards and Technology, Gaithersburg, MD, USA) was used to identify the by-products.

## 3. Results and Discussion

### 3.1. Structural and Morphological Characterization

The X-ray diffractometry (XRD) analysis of the CoFe_2_O_4_/Fe_2_O_3_-KIT-6 catalyst was carried out to identify whether the desired phases were achieved and to obtain information on the crystal structure of the CoFe_2_O_4_/Fe_2_O_3_-KIT-6 solid. The result is shown in Figure 1. The refined diffractogram is shown in Appendix A.

It was possible to identify, as shown in Figure 1A, the characteristic peaks of both the cobalt ferrite (JCPDS 01-079-1744) and hematite (JCPDS 01-087-1165) phases, which indicates that the mixed-phases CoFe_2_O_4_/Fe_2_O_3_ were obtained. The cobalt ferrite corresponds to 82 wt.% with a crystallite size of 21 nm, while hematite corresponds to 18 wt.% with a crystallite size of 26 nm.

The result presented in Figure 1B indicates that the long-range structural order was maintained after the synthesis steps, but there was a shift associated with the presence of the cobalt ferrite phase and hematite [29]. This shift of the peaks to higher angle observed for the supported material can be explained by the partial pore blockage concerning the mesoporous structure of KIT-6. The reduction of mesoporosity is due to the effect of partial pore filling by the incorporation of the cobalt/hematite ferrite phase in the support; however, the shift is due to the decrease in the spacing of the planes in the KIT-6 mesoporous structure and the presence of the impregnated material [30].

The textural properties were evaluated by N_2_ physisorption and the results are shown in Figure 2.

The two samples (Figure 2A) exhibited a type IV-a isotherm according to the IUPAC classification and type I hysteresis [31]. According to the literature, the combination of this type of isotherm and hysteresis is typical of ordered mesostructures with narrow mesopores distribution, which matches the expectation for the KIT-6 structure [32]. The isotherm of the catalyst at the partial pressure of 0.46 shows evidence of blocked or partially closed pores, which is common in mesoporous materials that were impregnated, which is consistent with the experiments performed.

The corresponding calculated textural parameters are listed in Table 1 for surface area (S_BET_), pore volume (V_p_), and pore diameter (D_p_).

The data referring to the textural properties presented in Table 1 showed that the surface area decreased after the impregnation of the Fe^3+^ and Co^2+^-based precursors. The pore volume also showed a slight decrease after the insertion of the Fe oxides. These results indicated that the metals inserted by impregnation are mainly on the surface of the catalyst and a small fraction of the pores were partially blocked by hematite and cobalt ferrite. Despite the change in textural properties after impregnation, the mesoporous structure of the KIT-6 support was preserved, corroborating the low-angle diffractograms.

Mössbauer spectroscopy was used to obtain information about the chemical environment of iron in the CoFe_2_O_4_/Fe_2_O_3_-KIT-6 catalyst. The experimental and fitting data are shown in Figure 3. The hyperfine parameters obtained after fitting the spectra are shown in Table 2.

The Mössbauer spectra for the catalyst recorded at 12 K and 300 K showed three sextets and one doublet, Figure 3A,B. The sextets are due to thermally blocked magnetic nanoparticles and the doublet is due to superparamagnetic nanoparticles. The sextets indicate the presence of Fe3+ in three different chemical environments due to hematite and cobalt ferrite. Thus, two sextets were assigned to iron ions in tetrahedral (sextet 1) and octahedral (sextet 2) positions [33,34]; the third sextet (sextet 3) was ascribed to the hematite phase. The doublet indicates the presence of nanoparticles with very small sizes and due to the Co-ferrite phase.

The relative absorption area of each component is proportional to the relative amount of the phases in the sample. Hematite has a minor phase of 8%; on the other side, Co-ferrite consisted of 92% of the whole sample. Bulk hematite was antiferromagnetic with a weak ferromagnetic behavior at 300 K [35]; however, nanometer size hematite particles usually show enhanced ferromagnetic behavior due to a large number of uncompensated Fe magnetic moments at the particle’s surface. The Co-ferrite is a ferrimagnetic material with Fe and Co ions occupying two sublattices (these sublattices have octahedral-O and tetrahedral-T sites, respectively); the magnetic moments (m) between these sublattices are antiparallel and the effective magnetic moment is given by the difference m_eff_ = m_O_ − m_T_. The superparamagnetic Co-ferrite nanoparticles have their magnetic moments, relaxing between the easy axis on the crystal structure; however, in the presence of a magnetic field, the moments undergo blocking. The sextets ascribed to Co-ferrite indicated stable magnetic moments that easily respond to an external magnetic field.

By analyzing these data, one can conclude that the catalyst consists of particles with a wide size distribution, where the smallest particles were in the superparamagnetic regime and the largest were thermally blocked [36,37]. The hyperfine parameters and the relative amount of phases in each chemical environment are shown in Table 2.

The isomer shift of all components are consistent with the high spin state of Fe^3+^. The relative absorption area of sextet 1 and sextet 2 reflected the iron distribution in the Co-ferrite structure; at 12 K, the areas for these sextets were a little bit different, indicating that the Fe has unequal occupancies at these sites. It indicates that the cobalt ferrite was in the mixed spinel state [38].

The transmission electron microscopy (TEM) analysis was performed to better visualize the morphology and particle size distribution. The result is depicted in Figure 4.

The particle size was determined using the program ImageJ; the observed particles have sizes between 15 and 40 nm, corroborating the crystallite size obtained by the Scherrer equation. The large amplitude in the particle size distribution analyzed in the TEM agrees with the found in the Mössbauer study. In pure KIT-6, the wall thickness and channels were 4.6 and 3.6 nm respectively. Furthermore, by analyzing Figure 4C, it was possible to determine the interplanar distance for the pure KIT-6 with an average value of 9.32 nm, in agreement with the value obtained by the low-angle XRD for the d(211) plane, Appendix A.

### 3.2. Chemical and Electronic Characterization

The acid-base properties of the solid CoFe_2_O_4_ -Fe_2_O_3_-KIT-6 were determined by CO_2_ temperature-programmed desorption of CO_2_ (TPD-CO_2_) and infrared with pyridine adsorption (FTIR-Py). The results are shown in Figure 5.

The basic sites present in a solid can be classified as weak when CO_2_ desorption occurs between 20 °C and 160 °C, medium when CO_2_ desorption occurs between 160 °C and 400 °C, and strong when CO_2_ desorption occurs above 400 °C [39]. Therefore, the temperature range where desorption occurs will indicate the basic strength of the material. The catalyst showed a main peak (Figure 5A) at a low temperature between 27 and 190 °C, which indicates the dominance of weak basic sites, which originate from the metal oxide and the KIT-6 mesoporous support [40]. In addition, the presence of a very broad and intense peak indicated a large number of weak basic sites. This temperature range also indicated that CO_2_ was preferentially adsorbed in a bicarbonate form [18].

By the FTIR-Py analysis, Figure 5B, bands were located at approximately 1450, 1490, and 1609 cm^−1^ in the solid CoFe_2_O_4_-Fe_2_O_3_ -KIT-6, and the bands of pure KIT-6 were at 1446.2 cm^−1^ and 1597.7 cm^−1^. The bands present in the KIT-6 can be related to the interactions of the silanol groups present in the structure with the pyridine molecule through hydrogen bonding [18]. These bands can no longer be noticed in the spectrum of the catalyst CoFe_2_O_4_-Fe_2_O_3_ -KIT-6 because the calcination process at 700 °C transformed the silanol groups into siloxanes.

Moreover, these bands can be related to available Lewis acid sites, such as the Fe^3+^ in the hematite phase and the Fe^3+^/Co^2+^ in the Co- ferrite phase. The bands detected at 1490 and 1609 cm^−1^ did not appear on the pure support, indicating that the Lewis acid sites originated mainly from the hematite and cobalt ferrite structures [41]. It was not possible to notice bands attributed to acidic Brønsted sites since metal oxides generally have only acidic Lewis sites.

From the reflectance study (Appendix A), the spectrum [F(R∞)hυ]^1/n^ vs hυ (h is the Planck constant and υ is the frequency) was obtained using the Tauc plot using by the Kubelka–Munk model, which is fundamental to determining the band gap energy. These plots are shown in Figure 6. Hence, it was considered as *n* = 2 for the allowed indirect transitions and *n* = 1/2 for direct transitions, which, according to the literature are predominant in iron-based oxides [25].

The band gap energies for the catalyst were equal to 1.72 eV and 1.66 eV for the direct and indirect cases, respectively. These values are close to the values found in the literature of 1.76 eV [20] and 2.4 eV for the cobalt ferrite-based phase [42,43], considering that Co ferrite is the predominant phase in the CoFe_2_O_4_ -Fe_2_O_3_-KIT-6 solid.

The support did not affect the band gap value, as shown in the absorbance spectra. Nevertheless, the lower value found for the pure ferrite may be due to the presence of hematite. The band gap value depends on the size and shape of particles; thus, the energy increased as the particle size of semiconductor nanomaterials decreased [44]. This may explain the different values found in the literature, especially for pure hematite, which has a typical value of 2.2 eV [45].

The CoFe_2_O_4_/Fe_2_O_3_-KIT-6 material had a low band gap value and good absorption in the ultraviolet and visible regions; these can be advantageous for photocatalytic processes, since more photons can be absorbed by the heterostructure, and, thus, more electron-hole pairs can be generated under the visible or ultraviolet light originating from sunlight or artificial light, which significantly increases the possibilities of applications of this material.

By analyzing the spectra, it was possible to notice that the main interactions of the sample with the electromagnetic radiation occurred in the regions from 200 to 300 nm and in the region that extends from 550 to 750 nm, with exception of the pure KIT-6, which exceeded the detection limits of the equipment. From the absorbance spectrum shown in Figure 6D, the absorption of the solid CoFe_2_O_4_-Fe_2_O_3_ KIT-6 and pure cobalt ferrite were much larger than pure hematite. The results also indicated that the presence of KIT-6 in the catalyst did not significantly interfere with the absorbance/reflectance, which shows that the main active site for photocatalytic reactions is the CoFe_2_O_4_/Fe_2_O_3_ phase. Furthermore, the sample CoFe_2_O_4_-Fe_2_O_3_ -KIT-6 showed good interaction with UV-Vis radiation, which also suggests that the sample is optically active in this region.

Zeta potential analysis was performed to evaluate the nature of the relative charges on the surface of the materials in aqueous solutions as the pH was varied, which are important parameters that indicate whether the interaction is repulsive or attractive during surface reactions. The results are shown in Figure 7.

The Zeta potential of the materials was very similar, showing good stability when pH changes, with a negative potential in a basic medium, and positive when the medium became acid, which is consistent with the characteristics presented in the literature for similar samples [46,47]. The deviations found are related to differences in particle size and the combination of phases. The isoelectric point (IEP) values for the cobalt ferrite, KIT-6, and the catalyst (CoFe_2_O_4_/Fe_2_O_3_-KIT-6) indicated the pH at which the zeta potential was equal to zero, occurring in the pH range of 4 and 5 [48].

The hysteresis curves obtained by VSM for both the CoFe_2_O_4_/Fe_2_O_3_-KIT-6 catalyst and pure CoFe_2_O_4_ showed ferromagnetic behavior. The saturation magnetization and coercive field values for the pure cobalt ferrite are in agreement with the literature [18]. The results are shown in Figure 8.

The magnetization of the supported photocatalyst (Figure 8B) was lower when compared with pure cobalt ferrite (Figure 8A). However, for the CoFe_2_O_4_/Fe_2_O_3_-KIT-6 catalyst, the lower magnetization of hematite and the presence of non-magnetic KIT-6 support justify the lower saturation magnetization found in the hysteresis curve. Additionally, the low magnetization is also related to the low concentration of ferrite present in the catalyst, which corresponds to less than 20% of the total mass [10,18]. Despite this reduction, the catalyst still showed strong interactions with magnets, enabling its magnetic separation from the reaction medium.

### 3.3. Application in Dye Degradation

Figure 9 shows the results of the photocatalytic tests performed with the ultra RGB red remazol dye (URRD) with the CoFe_2_O_4_/Fe_2_O_3_–KIT-6 catalyst. For the titanium oxide-modified solid containing anatase, the photocatalytic performance is shown in Appendix A. Appendix A shows UV-Vis spectra concerning the dye degradation by the CoFe_2_O_4_/Fe_2_O_3_-KIT-6 and TiO_2_/CoFe_2_O_4_/Fe_2_O_3_-KIT-6 catalysts, with and without H_2_O_2_.

The degradation rates in 1 h of reaction for the solids evaluated in the tests are shown in Table 3.

The CoFe_2_O_4_/Fe_2_O_3_-KIT-6 photocatalyst did not show (Figure 9A) considerable dye adsorption for 60 min, considering that the solid did not show significant changes in the initial concentration of ultra RGB red remazol in the absence of light. Appendix A demonstrates adsorption in the absence of light for 30 min before the photocatalytic tests, confirming that adsorption did not significantly contribute to the photocatalytic process.

The results indicated that the CoFe_2_O_4_/Fe_2_O_3_–KIT-6 catalyst is active and efficient in the degradation of the ultra RGB red remazol dye using a heterogeneous Photo-Fenton-type reaction, presenting degradation results equivalent to the commercial TiO_2_, which is a classic photocatalyst with good results in dye degradation. The low degradation in the absence of H_2_O_2_ indicated that the activity of CoFe_2_O_4_/Fe_2_O_3_–KIT-6 depends on the presence of H_2_O_2_. The Kinetic parameters for the dye degradation reaction are shown in Table 4.

The catalyst degraded the same percentage as the Photolysis/H_2_O_2_ test, even after 2 h of reaction. The degradation rate in the presence of hydrogen peroxide was more intense for the CoFe_2_O_4_/Fe_2_O_3_–KIT-6 solid. This fact is consistent with the mechanisms for the heterogeneous Photo-Fenton reaction, which demonstrates that the material is capable of providing a greater conversion of peroxide into hydroxyl radicals, even at lower H_2_O_2_ concentrations. It is important to highlight that the presence of the hematite phase (Fe_2_O_3_) did not significantly alter the degradation of the dye. The CoFe_2_O_4_/Fe_2_O_3_ sample in the absence of the KIT-6 support showed slightly lower degradation kinetics compared with the supported solid, Appendix A.

The combination of CoFe_2_O_4_/Fe_2_O_3_–KIT-6 material with TiO_2_ showed a significant improvement in the final activity of the material in a peroxide-free medium compared with the CoFe_2_O_4_/Fe_2_O_3_–KIT-6 catalyst without titanium, Appendix A. Furthermore, the photocatalysts TiO_2_/CoFe_2_O_4_/Fe_2_O_3_–KIT-6 and CoFe_2_O_4_/Fe_2_O_3_–KIT-6 in the presence of H_2_O_2_ showed the same degradation rate compared to the commercial TiO_2_; however, they can be removed using an external magnetic field, making these materials more advantageous for application in the treatment of effluents due to its ease of recovery. Furthermore, increasing the percentage of TiO_2_ can lead to an improvement in the activity of the TiO_2_/CoFe_2_O_4_/Fe_2_O_3_–KIT-6 material and reduce the need to use H_2_O_2_. It is important to highlight that the reaction using TiO_2_ dispersed in KIT-6, without cobalt ferrite, was slightly slower and less efficient compared with the CoFe_2_O_4_/Fe_2_O_3_–KIT-6 sample (Appendix A).

The R^2^ values presented in Table 4 and Figure 9B confirm that the reaction under study followed pseudo-first order kinetics due to the linear behavior of the ln (C_t_/C_0_) versus time. The values obtained for the rate constant and half-life (k` (min^−1^) and t_1/2_(min)) are in agreement with what was analyzed previously, where the tests with H_2_O_2_ showed better performance in the degradation of the dye and the addition of titanium in the structure improved the photocatalyst performance both in the presence and absence of hydrogen peroxide. The result also confirmed that the reaction was completed within 1 h of reaction for the photocatalyst/H_2_O_2_ systems. The kinetic parameters also confirmed that using only light is not enough to degrade the dye, presenting a low rate constant and a high half-life.

The material CoFe_2_O_4_/Fe_2_O_3_-KIT-6 has interesting magnetic properties, as observed in the magnetic results, which allows its easy recovery from the reaction medium and subsequent reuse in other photocatalytic tests as shown in the inset of Figure 8.

These results indicate that the material has an advantage compared with other solids used as photocatalysts/adsorbents, such as pure TiO_2_, which is difficult to separate from the reaction medium [22]. At the end of the photocatalytic test, it was possible to recover 85% of the original mass of the CoFe_2_O_4_/Fe_2_O_3_-KIT-6 photocatalyst.

Different recycling tests were performed for the sample CoFe_2_O_4_/Fe_2_O_3_-KIT-6, which was magnetically recovered after each reuse in order to evaluate the photocatalyst stability. The results are shown in Figure 10. The UV-VIS spectra regarding the dye degradation in each circle are presented in Appendix A.

The results indicated that CoFe_2_O_4_/Fe_2_O_3_-KIT-6 in the presence of H_2_O_2_ maintained its high efficiency in dye degradation even after recycling several times, suggesting that there was no destruction or inactivation of the main active phase concerning cobalt ferrite between the cycles. Magnetic separation was efficient for solid recovery at the end of each cycle, confirming the recyclability of this material compared with the pure anatase phase.

In addition, the X-ray diffractogram of the CoFe_2_O_4_/Fe_2_O_3_-KIT-6 catalyst recovered after reaction using magnetic separation is shown in Figure 11.

The diffractogram indicates that the main active phase, CoFe_2_O_4_, remained after the reaction process, which suggests that the solid can be reused in several reaction cycles. The decrease in the hematite amount peaks may indicate the loss of this phase by contact with the acid and hydrogen peroxide present in the reaction medium; however, further detailed studies is necessary to confirm this assumption. Despite the observed loss, the degradation process of the ultra RGB red remazol dye was affected, since the presence of the hematite phase (Fe_2_O_3_) was not significant in the process according to the recycle results presented in Figure 10.

### 3.4. Degradation Mechanism

The possible degradation mechanism for remazol ultra RGB red (URRD) with the catalyst studied was proposed and described through a sequence of reactions taking into account a Photo-Fenton system perspective. Initially the photocatalyst absorbed a photon with energy *h*v (*h* is the Planck constant and v is the frequency of the radiation) generating the electron/hole pairs (e^−^/h^+^) as shown in Reaction (1). When water interacted with the surface of the catalyst, it produced OH• radicals, which are highly oxidizing, while the interaction of the contaminant with the surface of the catalyst is capable of generating a radical from the first breakage in the dye molecule, according to reactions (2) and (3), respectively.


**Surface steps**

(1)
$${4\mathrm{hv}+\mathrm{CoFe}}_{2}O_{4}{-\mathrm{Fe}}_{2}O_{3}/\mathrm{KIT}-6\;\overset{\;\;\;\;\;\;\;\;\;\;\;\;\;\;\;\;\;}{⟶}{\;4h}^{+}{}_{\mathrm{VB}}+4e^{-}{}_{\mathrm{CB}}$$


(2)
                           2h+VB+2H2O ⟶                  2H++2O•H


(3)
$$\;\;\;\;\;\;\;\;\;\;\;\;\;\;\;\;\;\;\;\;\;\;\;\;\;\;\;\;\;2\mathrm{URRD}+2h^{+}{}_{\mathrm{VB}}\;\overset{\;\;\;\;\;\;\;\;\;\;\;\;\;\;\;\;\;\;}{⟶}{\;2\mathrm{URRD}}^{*}+2H^{+}$$




**Solution steps**

(4)
                             e−CB + H2O2 ⟶                   O−H + O•H


(5)
$$\;\;\;\;\;\;\;\;\;\;\;\;\;\;\;\;\;\;{2e}^{-}{}_{\mathrm{CB}}+2O_{2}\;\overset{\;\;\;\;\;\;\;\;\;\;\;\;\;\;\;\;\;\;\;}{⟶}{\;2O}_{2}{}^{-}\cdot$$


(6)
$${\;\;\;\;\;\;\;\;2H}^{+}+2O_{2}{}^{-}\cdot \overset{\;\;\;\;\;\;\;\;\;\;\;\;\;\;\;\;\;\;}{⟶}{\;2\mathrm{HO}}_{2}$$


(7)
$${\;\;\;\;\;\;\;\;\;\;\;\mathrm{HO}}_{2}\;+\;{\mathrm{HO}}_{2}\overset{\;\;\;\;\;\;\;\;\;\;\;\;\;\;\;\;\;\;}{⟶}{\;H}_{2}O_{2}{\;+\;O}_{2}$$


(8)
            e−CB+H2O2 ⟶                    O−H+O•H


(9)
$${\;\;\;\;\;\;\;2H}^{+}+2{\mathrm{OH}}^{-}\;\;\overset{\;\;\;\;\;\;\;\;\;\;\;\;\;\;\;\;\;\;\;\;}{⟶}\;2H_{2}O$$


(10)
      2URRD*+ 4O•H ⟶                     H2O+CO2




**Global Reaction**

(11)
$$4\mathrm{hv}\;+\;2{\mathrm{URRD}\;+\;H}_{2}O_{2}{\;+\;O}_{2}\overset{{\mathrm{CoFe}}_{2}O_{4}{-\mathrm{Fe}}_{2}O_{3}/\mathrm{KIT}-6}{⟶}H_{2}{O\;+\;\mathrm{CO}}_{2}$$



The generated electron was capable of promoting a series of chain reactions. First, the interaction of hydrogen peroxide added in the Photon-Fenton processes also produced the hydroxyl radicals, Reaction (4). Oxygen is another molecule capable of interacting with the electron, generating the O_2_-• radical, Reaction (5), which, in the presence of H^+^, resulting from both the acidity of the reaction medium and from Reaction (3), produced HO_2_• radicals, Reaction (6). This is followed by producing new molecules of H_2_O_2_, as exposed in the Reaction (7). Subsequently, in Reaction (8), the peroxide underwent the same reaction proposed in Reaction (4).

In Reaction (9), the H^+^ and OH^-^ ions generated during the reactions combined to generate water molecules. The last mechanism step, Reaction (10), demonstrates the mineralization of the URRD* radical formed in Reaction (2) by undergoing oxidation in the presence of hydroxyl radicals produced during the process. Finally, the overall reaction (Equation (11)) summarizes all these steps, eliminating the species generated in the reaction medium and demonstrating the conditions and reactants necessary in the presence of the catalyst, CoFe_2_O_4_/Fe_2_O_3_-KIT-6, to promote the mineralization of the URRD dye.

To illustrate this sequence of reactions, Figure 1 is presented, where one can see the production of electron/hole pairs in the conduction and valence bands, respectively, when the catalyst had its surface irradiated by a source of light energy (hv) as demonstrated in Reaction (1). It is also possible to visualize the production of OH•, O_2_-•, and HO_2_• radicals as occurring in Reactions (2) to (9) and the effect of these radicals on the degradation and mineralization of the URRD dye, according to Reaction (10) and the overall Reaction (11).

It is important to mention that the sequence of reactions can lead to the formation of different by-products before the complete dye degradation. The GC-MS results indicated the formation of possible by-products such as 1-Methyl-3-phenylindole and 2,4-Dimethylbenzo[h]quinolone for the reaction using the CoFe_2_O_4_/Fe_2_O_3_-KIT-6 catalyst in the presence of H_2_O_2_. The results are presented in Appendix A.

## 4. Conclusions

It was possible to confirm the efficiency of the impregnation of cobalt ferrite and hematite on KIT-6. The KIT-6 support did not undergo significant changes. In addition, metal oxide particles with varying sizes were observed in the TEM study. The presence of Lewis acid and weak basic sites were confirmed. The sample exhibited low band-gap energy and easy magnetic separation, indicating great potential for application as a reusable photocatalyst.

It was confirmed that the photocatalyst CoFe_2_O_4_/Fe_2_O_3_–KIT-6 is a promising material for photodegradation of the ultra RGB red dye in the presence of H_2_O_2_ using a Photo-Fenton reaction by obtaining a maximum degradation rate after 60 min of reaction. The impregnation of the material CoFe_2_O_4_/Fe_2_O_3_–KIT-6 with TiO_2_ (anatase) proved to be efficient, showing good degradation results for applications in absence of H_2_O_2_. All proposed catalysts, within the heterogeneous Photo-Fenton system, showed the same degradation rate compared to commercial TiO_2_; however, the CoFe_2_O_4_/Fe_2_O_3_–KIT-6 sample has the advantage of easy magnetic separation.

The proposed mechanism indicated that the dye underwent a sequence of elemental radical reactions on the surface and in solution to produce oxidant radicals such as OH•, which are essential for the complete mineralization of the contaminating matrix.

## Data Availability

Not applicable.

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
