# Peer review of "Synthesis, Characterization and Photocatalytic Activity of CoFe2O4/Fe2O3 Dispersed in Mesoporous KIT-6"

_nanomaterials, 2022, doi:10.3390/nano12203566_

Round 1

Reviewer 1 Report

The samples preparation was well explained. The characterization of samples using different techniques it was also extensively explained and correlated.

However, some problems should be addressed before this paper can be considered to be accepted.

 1 - The English writing should be improved.

 2 – Reuse experiments of the catalyst must be performed.

 3 - The by-products produced during the photodegradation should be analyzed.

4 - Photocatalytic efficiency comparison of CoFe2O4/Fe2O3 with and without KIT-6 is missing.

5 - The results of the adsorption in the absence of light during 30 min, have to appear in the manuscript.

I think that the results can be published, however, these concerns should be provided first in a revised form.

Author Response

Dear,

We are submitting for your consideration, a manuscript entitled “SYNTHESIS, CHARACTERIZATION AND PHOTOCATALYTIC ACTIVITY OF CoFe2O4/Fe2O3 DISPERSED IN MESOPOROUS KIT-6”, with the aim of publishing as an original research study, in the nanomaterials journal, Special Issue Title: Synthesis and Application of Silicon Dioxide Nanoparticles. Some clarifying modifications were made and a major revision of indicated points were performed. The required points are presented below according to the request. The changes in the text were highlighted in the manuscript.

  1. The English writing should be improved.

Answer:

The manuscript has been revised to improve English quality. Thanks for the suggestion.

  1. Reuse experiments of the catalyst must be performed.

Answer:

This is an interesting suggestion. Four reuses were performed after the initial test for the photocatalyst CoFe2O4/Fe2O3-KIT-6 in the presence of H2O2, which showed that the catalyst did not lose efficiency over the cycles. The results are shown in Figure 10 and S7. The information was described in the manuscript.

  1. The by-products produced during the photodegradation should be analyzed.

Answer:

This is also an interesting suggestion. An initial analysis of GC-Ms was performed which was identified the possible major by-products for the photodegradation of the URRD dye and are shown in Table S3. The analyzed sample was collected in 15 min of reaction using the photocatalyst CoFe2O4/Fe2O3-KIT-6 in the presence of H2O2. Due to the short period of time for review, more robust studies varying the analysis conditions will be carried out in future works. We count on your understanding.

  1. 4. Photocatalytic efficiency comparison of CoFe2O4/Fe2O3 with and without KIT-6 is missing.

Answer:

Indeed, we did not enter the result of the unsupported CoFe2O4-Fe2O3 sample. We enter the results in Figure S6-F. Dye degradation kinetics were worse for the solid unsupported sample in KIT-6, confirming the effect of support on catalytic performance.

  1. 5. The results of the adsorption in the absence of light during 30 min, have to appear in the manuscript.

Answer:

              This is really an interesting result to insert in the manuscript. These data were added to Figure S3 referring to the 30 min adsorption in the absence of light, prior to the start of the photocatalytic reaction. This result can also be seen in the black line in Figure 9A.

Thank you in advance for your kind attention and suggestions. We are truly interested in publishing this article in the nanomaterials journal, Special Issue Title: Synthesis and Application of Silicon Dioxide Nanoparticles. We agree that the article needed a major revision before publication. We modified several points in the text and after modifications; we believe that is sufficient for publication in the nanomaterials journal and can contribute to the literature on this topic.

We are looking forward to hearing from you.

Sincerely,

Reviewer 2 Report

The authors synthesized a CoFe2O4/Fe2O3 catalyst supported on mesoporous KIT-6, made detailed characterizations and finally applied it on photocatalytic dye degradation. The manuscript should be improved by addressing the following issues before making further consideration:

1.     For all the figures with intensity as y axis, the unit should be (a.u.) not (u.a). The x axis of XRD should be 2θ (degree).

2.     In Figure 4, what is the difference between 4A and 4B? Can you show some detailed information of CoFe2O4 or Fe2O3 from TEM?

3.     For photocatalytic dye degradation, the CoFe2O4/Fe2O3–KIT-6 catalyst only worked in presence of H2O2, however, TiO2 can work efficiently in absence of H2O2, can you explain the advantage of the CoFe2O4/Fe2O3–KIT-6 catalyst over TiO2 and the working mechanism of these two photocatalysts?

4.     In Figure S4, the authors should add a comparison with TiO2/H2O2.

5.     The abstract and conclusion parts should be concentrated and give an overview of the whole article, the authors talked too much details which have been displayed in the main text.

Author Response

Dear,

We are submitting for your consideration, a manuscript entitled “SYNTHESIS, CHARACTERIZATION AND PHOTOCATALYTIC ACTIVITY OF CoFe2O4/Fe2O3 DISPERSED IN MESOPOROUS KIT-6”, with the aim of publishing as an original research study, in the nanomaterials journal, Special Issue Title: Synthesis and Application of Silicon Dioxide Nanoparticles. Some clarifying modifications were made and a major revision of indicated points were performed. The required points are presented below according to the request. The changes in the text were highlighted in the manuscript.

  1. For all the figures with intensity as y axis, the unit should be (a.u.) not (u.a). The x axis of XRD should be 2θ (degree).

Answer:

The correction was made in XRD Figure. Thanks for the observation.

2.In Figure 4, what is the difference between 4A and 4B? Can you show some detailed information of CoFe2O4 or Fe2O3 from TEM?

Answer:

Unfortunately, from our TEM images, we can only get the particle size distribution as entered in the manuscript. It will not be possible to enter the electron diffraction, SAED results, which it would be possible to observe the plane distances of the CoFe2O4 and Fe2O3 phases and correlate with the XRD results. We are committed to carrying out more detailed analysis in future work. We count on your understanding.

  1. For photocatalytic dye degradation, the CoFe2O4/Fe2O3–KIT-6 catalyst only worked in presence of H2O2, however, TiO2 can work efficiently in absence of H2O2, can you explain the advantage of the CoFe2O4/Fe2O3–KIT-6 catalyst over TiO2 and the working mechanism of these two photocatalysts?

Answer:

              The photocatalysts proposed in the article can be removed with the aid of an external magnetic field gradient, making the materials more advantageous for application in the treatment of effluents due to their ease of recovery, according to the reuse results and the image inserted in the curves of magnetization. On the other hand, solids containing TiO2 without the magnetic phase based on CoFe2O4 are very difficult to separate, making it difficult to reuse them in different cycles. Thus, even if the sample containing pure TiO2 is slightly better, its application in different reuses is unfeasible. This information was written more clearly in the manuscript.

  1. 4. In Figure S4, the authors should add a comparison with TiO2/H2O2.

Answer:

This is an interesting suggestion. A study of TiO2/KIT-6 was carried out in the presence of H2O2, without the CoFe2O4-Fe2O3 phases, and the results indicate lower activity concerning dye degradation as seen in Figure S6-E, showing a lower rate compared to the CoFe2O4-Fe2O3-KIT-6.

  1. 5. The abstract and conclusion parts should be concentrated and give an overview of the whole article, the authors talked too much details which have been displayed in the main text.

Answer:

Indeed, some information in the abstract and conclusion was repetitive compared to the main text. We have eliminated and rewritten some parts in order to improve the quality of the text, emphasizing general information. It is important to note that the number of characters in the abstract is very small, making detailing difficult.

Thank you in advance for your kind attention and suggestions. We are truly interested in publishing this article in the nanomaterials journal, Special Issue Title: Synthesis and Application of Silicon Dioxide Nanoparticles. We agree that the article needed a major revision before publication. We modified several points in the text and after modifications; we believe that is sufficient for publication in the nanomaterials journal and can contribute to the literature on this topic.

We are looking forward to hearing from you.

Sincerely,

Reviewer 3 Report

The manuscript “SYNTHESIS, CHARACTERIZATION AND PHOTOCATALYTIC ACTIVITY OF CoFe2O4/Fe2O3 DISPERSED IN MESO-POROUS KIT-6” reports the synthesis of a hybrid structure formed by a porous template in which CoFe2O4/Fe2O3 nanostructures were dispersed. Further, anatase was added in order to improve the photo-degradation activity of the composite material toward the Remazol red ultra RGB dye.

The catalyst was characterized by several techniques and the proposed rationales about phase, structure and morphology of the synthetized material are supported by the experimental data.

The performances of the proposed catalyst are, in my opinion, far for proposing it as an innovative and efficient photocatalyst. In fact, as reported in Table 3, commercial TiO2 photocatalyst performances are similar enough to the CoFe2O4/Fe2O3-Kit-6 in presence of H2O2 and the high degradation rate obtained by means of photolysis/H2O2 further confirms that the proposed system (containing the environmental dangerous Cobalt) is not competitive as a performing catalyst.

Other critical points should be solved by the authors. For example, they reported the XRD of CoFe2O4/Fe2O3-Kit-6 catalyst after the degradation reaction and, as reported, it appears different from the as-synthetized materials. Nevertheless, the authors didn’t check the reuse of materials as photo-catalyst on the dye.

Further, important information is missing in the manuscript: the UV-Vis spectrum of the decomposed dye was never reported, the description of the light source used for photodegradation the materials was not detailed (is it a solar simulator, a tungsten lamp, a halogen one?).

As minor points:

·       In figure 8A is reported CoFe2O4/Fe2O3-KIT-6/Adsorption but it is discussed neither in the manuscript text nor in the figure caption

·       In table 4 kinetic parameters related to photolysis are missing.

Author Response

Dear referee,

We are re-submitting for your consideration, a manuscript entitled “SYNTHESIS, CHARACTERIZATION AND PHOTOCATALYTIC ACTIVITY OF CoFe2O4/Fe2O3 DISPERSED IN MESOPOROUS KIT-6”, with the aim of publishing as an original research study, in the nanomaterials journal, Special Issue Title: Synthesis and Application of Silicon Dioxide Nanoparticles. Some clarifying modifications were made and a major revision of indicated points were performed. The required points are presented below according to the request. The changes in the text were highlighted in the manuscript.

  1. The performances of the proposed catalyst are, in my opinion, far for proposing it as an innovative and efficient photocatalyst. In fact, as reported in Table 3, commercial TiO2 photocatalyst performances are similar enough to the CoFe2O4/Fe2O3-Kit-6 in presence of H2O2 and the high degradation rate obtained by means of photolysis/H2O2 further confirms that the proposed system (containing the environmental dangerous Cobalt) is not competitive as a performing catalyst.

Answer:

This is comment to be pondered. In fact, the catalyst containing pure TiO2 was similar to the catalyst containing ferrite. However, according to our experience with samples based on TiO2, they are very difficult to separate at the end of the process, even by filtration process, which makes their reuse by successive tests unfeasible. On the other hand, photocatalysts containing ferrites are easily separated magnetically, facilitating their reuse and long-term use, thus, showing advantages compared to pure TiO2.

This information was better described in the manuscript.

  1. Other critical points should be solved by the authors. For example, they reported the XRD of CoFe2O4/Fe2O3-Kit-6 catalyst after the degradation reaction and, as reported, it appears different from the as-synthetized materials. Nevertheless, the authors didn’t check the reuse of materials as photo-catalyst on the dye.

Answer:

This was an important suggestion to improve the quality of the manuscript.

Four reuses were performed after the initial test for the photocatalyst CoFe2O4/Fe2O3-KIT-6 in the presence of H2O2, which showed that the catalyst did not lose efficiency over the cycles.

The results confirmed the photocatalyst stability, even though it was not possible to observe the Fe2O3 phase after the reaction according to the XRD results of the post-reaction solids, confirming that the CoF2O4 phase is the main active site. Such information was better described in the manuscript.

  1. Further, important information is missing in the manuscript: the UV-Vis spectrum of the decomposed dye was never reported.

Answer:

              Indeed, it is interesting to insert the different spectra at different times. Spectra were added in Figure S6 for the main catalysts.

  1. 4. The description of the light source used for photodegradation the materials was not detailed (is it a solar simulator, a tungsten lamp, a halogen one?).

Answer:

The possible information was described in the methodology. A 125 W mercury vapor lamp was used.

  1. 5. In figure 8A is reported CoFe2O4/Fe2O3-KIT-6/Adsorption but it is discussed neither in the manuscript text nor in the figure caption.

Answer:

The adsorption values are really indispensable. The information was better described in the manuscript.

  1. In table 4 kinetic parameters related to photolysis are missing.

Answer:

This is also an interesting and necessary suggestion. The values were entered in the Table.

Thank you in advance for your kind attention and suggestions. We are truly interested in publishing this article in the nanomaterials journal, Special Issue Title: Synthesis and Application of Silicon Dioxide Nanoparticles. We agree that the article needed a major revision before publication. We modified several points in the text and after modifications; we believe that is sufficient for publication in the nanomaterials journal and can contribute to the literature on this topic.

We are looking forward to hearing from you.

Sincerely,

Round 2

Reviewer 1 Report

The authors have answered all the comments in detail and revised the paper according to the comments and suggestions to improve it.

However, some mistakes remain in the manuscript, for example in n table 3 it is not clear which tests were performed with and without H2O2. The legend and sample name must be maintained and consistent throughout the manuscript.

Furthermore, the performance of the CoFe2O4/Fe2O3-Kit-6, the new catalyst proposes by the authors, it is not good enough when compared with the TiO2 commercial (for example Fig. S5 and Fig. 9).

In Fig. S5 the performance of the TiO2 and TiO2/CoFe2O4/Fe2O3-Kit-6 is very similar.

In Fig. S6, when compared the TiO2/CoFe2O4/Fe2O3-Kit-6 and CoFe2O4/Fe2O3-Kit-6 (Fig. S6A and S6D), the sample with TiO2 showed better catalytic performance. The same is observed when compared the Fig. S6B and Fig. S6C.

And the Fig. S6A and S6F showed that the presence of KIT-6 don’t improve the performance of the materials.

So, the new catalyst that the authors propose in this work it is not promising when compared with the TiO2, more efforts should be performed to get better results. For example, the manuscript modification in order to show that the combination of the CoFe2O4/Fe2O3-KIT-6 with TiO2 is an advantage. Otherwise, the manuscript as it is structured makes no sense. Since when comparing the CoFe2O4/Fe2O3-KIT-6 catalytic activity with TiO2, it does not present significant improvement.

Author Response

Dear,

We are re-submitting for your consideration, a manuscript entitled “SYNTHESIS, CHARACTERIZATION AND PHOTOCATALYTIC ACTIVITY OF CoFe2O4/Fe2O3 DISPERSED IN MESOPOROUS KIT-6”, with the aim of publishing as an original research study, in the nanomaterials journal, Special Issue Title: Synthesis and Application of Silicon Dioxide Nanoparticles. Some clarifying modifications were made and a major revision of indicated points were performed. The required points are presented below according to the request. The changes in the text were highlighted in the manuscript.

  1. Some mistakes remain in the manuscript, for example in table 3 it is not clear which tests were performed with and without H2O2. The legend and sample name must be maintained and consistent throughout the manuscript.

Answer:

The reviewer is correct, we modified Table 2 as well as Tables 3 and S2 specifying whether the reactions were done with or without H2O2. Thanks for the suggestion.

  1. Furthermore, the performance of the CoFe2O4/Fe2O3-Kit-6, the new catalyst proposes by the authors, it is not good enough when compared with the TiO2 commercial (for example Fig. S5 and Fig. 9).

3-In Fig. S5 the performance of the TiO2 and TiO2/CoFe2O4/Fe2O3-Kit-6 is very similar.

4-In Fig. S6, when compared the TiO2/CoFe2O4/Fe2O3-Kit-6 and CoFe2O4/Fe2O3-Kit-6 (Fig. S6A and S6D), the sample with TiO2 showed better catalytic performance. The same is observed when compared the Fig. S6B and Fig. S6C.

Answer:

The photocatalysts proposed in the article can be removed with the aid of an external magnetic field gradient, making the materials more advantageous for application in the treatment of effluents due to their ease of recovery, according to the reuse results and the image inserted in the curves of magnetization. On the other hand, solids containing TiO2 without the magnetic phase based on CoFe2O4 are very difficult to separate, making it difficult to reuse them in different cycles. We recover only 30-40% of the TiO2 solid in the first recycle, while it is possible to recover 90-95% of the solid CoFe2O4/Fe2O3-KIT-6 at each reuse. Thus, even if the sample containing pure TiO2 is slightly better, its application in different reuses is unfeasible. This information was written more clearly in the manuscript.

5-And the Fig. S6A and S6F showed that the presence of KIT-6 don’t improve the performance of the materials.

Indeed, the degradation kinetics of the sample containing KIT-6 was similar to the unsupported solid, however, taking into account that KIT-6 provides a better dispersion of the active sites based on cobalt ferrite and a better porosity, the solids containing KIT-6 can resist photocatalytic tests for long periods of time, considering that it is more resistant to blocking pores. This information has already been described in the introduction.

  1. So, the new catalyst that the authors propose in this work it is not promising when compared with the TiO2, more efforts should be performed to get better results. For example, the manuscript modification in order to show that the combination of the CoFe2O4/Fe2O3-KIT-6 with TiO2 is an advantage. Otherwise, the manuscript as it is structured makes no sense. Since when comparing the CoFe2O4/Fe2O3-KIT-6 catalytic activity with TiO2, it does not present significant improvement.

Answer: The advantage of the proposed catalyst (CoFe2O4/Fe2O3-KIT-6) compared to the TiO2 commercial catalyst has already been mentioned in the above answer and also described in the manuscript.

Concerning the TiO2-CoFe2O4/Fe2O3-KIT-6 photocatalyst, it showed a slight improvement compared to solid CoFe2O4/Fe2O3-KIT-6 in the absence of H2O2. It is important to mention that, for this sample, it was just an initial study, thus, we put these results in the supplemental materials. Future studies will be carried out using different TiO2 contents on CoFe2O4/Fe2O3-KIT-6 sample to improve its catalytic performance. This is just an initial study that will serve as a basis for further studies.

Regardless, we believe that both catalysts (CoFe2O4/Fe2O3-KIT-6 and TiO2-CoFe2O4/Fe2O3-KIT-6) are advantageous compared to commercial TiO2 due to their magnetic properties, ease of separation and reuse.

             Thank you in advance for your kind attention and suggestions. We are truly interested in publishing this article in the nanomaterials journal, Special Issue Title: Synthesis and Application of Silicon Dioxide Nanoparticles. We modified several points in the text and after modifications; we believe that is sufficient for initial publication in the nanomaterials journal and can contribute to the literature on this topic.

We are looking forward to hearing from you.

Sincerely,

Reviewer 2 Report

I am satisfied with the changes that the authors made according to my comments.

Author Response

Thanks for the suggestions and the acceptance of the manuscript.

Reviewer 3 Report

The authors modified the manuscript and, undoubtedly, the scientific quality of the entire contribute appears substantially improved. Nevertheless, it is my opinion that the manuscript is not suitable yet for Nanomaterials but it could be resubmitted on more specialistic Journals.

Author Response

Dear referee,

We are re-submitting for your consideration, a manuscript entitled “SYNTHESIS, CHARACTERIZATION AND PHOTOCATALYTIC ACTIVITY OF CoFe2O4/Fe2O3 DISPERSED IN MESOPOROUS KIT-6”, with the aim of publishing as an original research study, in the nanomaterials journal, Special Issue Title: Synthesis and Application of Silicon Dioxide Nanoparticles. Some clarifying modifications were made and a major revision of indicated points were performed. The required points are presented below according to the request. The changes in the text were highlighted in the manuscript.

  1. The authors modified the manuscript and, undoubtedly, the scientific quality of the entire contribute appears substantially improved. Nevertheless, it is my opinion that the manuscript is not suitable yet for Nanomaterials but it could be resubmitted on more specialistic Journals.

Answer:

As mentioned in the comment above we have greatly improved the quality of the manuscript after the major revision.

We believe that the manuscript is suitable for the scope of the journal nanomaterials, since a complete characterization of the synthesized materials was performed and showed the formation of nanoparticles by XRD and TEM with possible application in various research fields such as photocatalysis. Furthermore, the special issue is related to the synthesis of silica-based nanomaterials. We present the synthesis of a nanomaterial containing KIT-6 (mesoporous silica), which is a very promising material based on silica and fits perfectly in the theme proposed by the nanomaterials journal.

 Thank you in advance for your kind attention and suggestions. We are truly interested in publishing this article in the nanomaterials journal, Special Issue Title: Synthesis and Application of Silicon Dioxide Nanoparticles. We agree that the article needed a major revision before publication. We modified several points in the text and after modifications; we believe that is sufficient for initial publication in the nanomaterials journal and can contribute to the literature on this topic.

We are looking forward to hearing from you.

Sincerely,

Round 3

Reviewer 1 Report

The authors have answered all the comments in detail and revised the paper according to the comments and suggestions to improve the manuscript. In my opinion the results can be published in the Nanomaterials journal. Therefore, I suggest the paper to be accepted.

Author Response

Dear,

Thank you for the important suggestions and for the acceptance.

We already revised the English in the previous version and we confirmed it again in this third step.

We will be available to make minor adjustments in the references at the proof stage, if necessary.

Best regards,